# Definition of the Prognostic Role of MGMT Promoter Methylation Value by Pyrosequencing in Newly Diagnosed IDH Wild-Type Glioblastoma Patients Treated with Radiochemotherapy: A Large Multicenter Study

**DOI:** 10.3390/cancers14102425

**Published:** 2022-05-13

**Authors:** Mario Caccese, Matteo Simonelli, Veronica Villani, Simona Rizzato, Tamara Ius, Francesco Pasqualetti, Marco Russo, Roberta Rudà, Rosina Amoroso, Luisa Bellu, Roberta Bertorelle, Francesco Cavallin, Angelo Dipasquale, Mariantonia Carosi, Stefano Pizzolitto, Daniela Cesselli, Pasquale Persico, Beatrice Casini, Matteo Fassan, Vittorina Zagonel, Giuseppe Lombardi

**Affiliations:** 1Department of Oncology, Oncology 1, Veneto Institute of Oncology IOV-IRCCS, 35128 Padua, Italy; vittorina.zagonel@iov.veneto.it (V.Z.); giuseppe.lombardi@iov.veneto.it (G.L.); 2Department of Biomedical Sciences, Humanitas University, Pieve Emanuele, 20090 Milan, Italy; matteo.simonelli@hunimed.eu (M.S.); angelo.dipasquale@cancercenter.humanitas.it (A.D.); pasquale.persico@hunimed.eu (P.P.); 3IRCCS Humanitas Research Hospital, 20089 Rozzano, Italy; 4Neuro-Oncology Unit, Regina Elena National Cancer Institute, 00161 Rome, Italy; veronica.villani@ifo.gov.it; 5Department of Oncology, Central Friuli University Health Authority, 33100 Udine, Italy; simona.rizzato@asufc.sanita.fvg.it; 6Neurosurgery Unit, Department of Neurosciences, Santa Maria della Misericordia University Hospital, 33100 Udine, Italy; tamara.ius@asufc.sanita.fvg.it; 7Radiation Oncology Unit, Pisa University Hospital, 56121 Pisa, Italy; f.pasqualetti@ao-pisa.toscana.it; 8Department of Oncology, University of Oxford, Oxford OX1 4BH, UK; 9Neurology Unit, Neuromotor Department, Azienda USL-IRCCS Reggio Emilia, 42121 Emilia, Italy; marco.russo@ausl.re.it; 10Department of Neuro-Oncology, University of Turin and City of Health and Science Hospital, 10094 Torino, Italy; roberta.ruda@aulss2.veneto.it; 11Neurology Unit, Hospital of Castelfranco Veneto, 31033 Castelfranco Veneto, Italy; 12Neurosurgery Unit, Department of Surgery, Hospital of Livorno, Azienda Asl Toscana Nord Ovest, 57100 Livorno, Italy; rosinamoroso@libero.it; 13Radiotherapy and Radiosurgery Department, IRCCS Humanitas Research Hospital, 20089 Rozzano, Italy; luisa.bellu@humanitas.it; 14Immunology and Molecular Oncology Unit, Department of Oncology, Veneto Institute of Oncology IOV IRCCS, 35128 Padua, Italy; roberta.bertorelle@iov.veneto.it; 15Independent Statistician, 36020 Solagna, Italy; cescocava@libero.it; 16Pathology Unit, Regina Elena National Cancer Institute, 00161 Rome, Italy; mariantonia.carosi@ifo.gov.it (M.C.); beatrice.casini@ifo.gov.it (B.C.); 17Department of Surgical Pathology, Central Friuli University Health Authority, 33100 Udine, Italy; stefano.pizzolitto@asufc.sanita.fvg.it; 18Department of Laboratory Medicine, Institute of Pathology, Santa Maria della Misericordia University Hospital, 33100 Udine, Italy; daniela.cesselli@uniud.it; 19Department of Medicine, University of Udine, 33100 Udine, Italy; 20Department of Oncology, Veneto Institute of Oncology, IOV-IRCCS, 35128 Padua, Italy; matteo.fassan@unipd.it; 21Cytopathology Unit, Department of Medicine (DIMED), Surgical Pathology & AMP, University of Padua, 35128 Padua, Italy

**Keywords:** glioblastoma, *MGMT*, methylation, temozolomide, pyrosequencing

## Abstract

**Simple Summary:**

O^6^-methylguanine (O^6^-MeG)-DNA methyltransferase (*MGMT*) methylation status is a predictive factor for alkylating treatment efficacy and glioblastoma patients, and its prognostic role is still unclear. The quantitative pyrosequencing (PSQ) approach has proven to be feasible for *MGMT* promoter methylation testing. We report the data of a large study analyzing *MGMT* promoter methylation status by pyrosequencing and its association with overall survival. We enrolled a large cohort of newly diagnosed Isocitrate Dehydrogenase (*IDH)* wild-type glioblastoma patients, all with rather homogeneous clinical characteristics, from nine neuro-oncology centers. Our data showed better survival when *MGMT* > 15% but also suggested a more complex (i.e., non-linear) relationship between survival and *MGMT*, resulting in an increase in the negative prognostic effect of a decrease in methylation of the *MGMT* promoter.

**Abstract:**

Background. O^6^-methylguanine (O^6^-MeG)-DNA methyltransferase *(MGMT)* methylation status is a predictive factor for alkylating treatment efficacy in glioblastoma patients, but its prognostic role is still unclear. We performed a large, multicenter study to evaluate the association between *MGMT* methylation value and survival. Methods. We evaluated glioblastoma patients with an assessment of *MGMT* methylation status by pyrosequencing from nine Italian centers. The inclusion criteria were histological diagnosis of *IDH* wild-type glioblastoma, Eastern Cooperative Oncology Group Performance Status (ECOG-PS) ≤2, and radio-chemotherapy treatment with temozolomide. The relationship between OS and *MGMT* was investigated with a time-dependent Receiver Operating Characteristics (ROC) curve and Cox regression models. Results. In total, 591 newly diagnosed glioblastoma patients were analyzed. The median OS was 16.2 months. The ROC analysis suggested a cut-off of 15% for MGMT methylation. The 2-year Overall Survival (OS) was 18.3% and 51.8% for *MGMT* methylation <15% and ≥15% (*p* < 0.0001). In the multivariable analysis, *MGMT* methylation <15% was associated with impaired survival (*p* < 0.00001). However, we also found a non-linear association between *MGMT* methylation and OS (*p* = 0.002): median OS was 14.8 months for *MGMT* in 0–4%, 18.9 months for *MGMT* in 4–40%, and 29.9 months for *MGMT* in 40–100%. Conclusions. Our findings suggested a non-linear relationship between OS and *MGMT* promoter methylation, which implies a varying magnitude of prognostic effect across values of *MGMT* promoter methylation by pyrosequencing in newly diagnosed *IDH* wild-type glioblastoma patients treated with chemoradiotherapy.

## 1. Introduction

Glioblastoma (GBM) represents the most common type of primary malignant brain tumor [1]. The clinical presentation and symptoms associated with this cancer can be as varied as headache, seizures, and motor and/or sensory neurological deficits.

The standard of care for patients with newly diagnosed GBM includes maximal safe surgery resection, followed by concurrent chemoradiotherapy treatment and subsequent chemotherapy with maintenance temozolomide (Stupp Protocol) [2]; this treatment involves the delivery of radiotherapy (60grays in 30 fractions) associated with chemotherapy treatment with temozolomide at a dosage of 75mg/m^2^ for the entire duration of the radiation treatment followed by chemotherapy alone with temozolomide at a dosage of 150–200 mg/m^2^ for 5 days every 28 days. The median overall survival ranges between 15 and 18 months [3,4] with a 5-year survival of less than 7% [5]. O^6^-methylguanine (O^6^-MeG)-DNA methyltransferase (MGMT) is a DNA repair protein that catalyzes the transfer of a methyl group from the guanine DNA nucleotide in O6-position to a cysteine residue at its own position 145 [6] and plays an important role in maintaining the integrity of the genome, and in repairing discrepancies and errors during DNA replication and transcription. This repair mechanism prevents cell death, gene mutations, and tumor transformation from occurring due to alkylating agents. 

Methylation of the MGMT gene promoter resulting in gene silencing results in low expression of the MGMT protein and reduced DNA repair activity, increasing sensitivity to alkylating agents, such as temozolomide [7,8]. The “EORTC 26981/22981-NCIC CE.3” study [9] demonstrated that MGMT promoter methylation can be predictor of temozolomide efficacy. The *MGMT* gene is located on chromosome band 10q26. It is composed of five exons and has an associated 5′-CpG island of 762 bp including 98 CpG dinucleotides [10] that encompasses a large part of the promoter and the first exon. In the absence of methylation of the CpG islands, the transcription sites of MGMT together with structures similar to nucleosomes (nucleosome-like) determine the transcription of this gene regularly. In the case of methylation of the CpGs islands, heterochromatinization is determined and does not allow for correct adherence of the nucleosome-like structure, masking the transcription initiation sites, making transcription impossible. Epigenetic alterations, such as the methylation of CpG islands, have always been considered early and frequent events in different types of cancer, playing a fundamental role in tumorigenesis and tumor progression. In fact, methylation usually determines the lack of mRNA, with a consequential loss of the protein and its enzymatic activity, often favoring tumor development.

At the moment, it is not yet clear which and how many CpGs in the MGMT CpG island should be methylated to determine a gene silencing in cancer cells to have an impact on treatment outcome and patient survival. Furthermore, all available diagnostic tests, whether qualitative, quantitative, or semi-quantitative, interrogate distinct sets of CpGs within the MGMT CpG island [11].

Methylation Specific PCR (MSP) and Pyrosequencing (PSQ) are the techniques most commonly used to detect the MGMT gene promoter methylation status. However, the use of the MSP is constrained by several problems, partly specific and in part common to the PSQ technique: the specificity and sensitivity of the method [12,13], the best performance of the technique if high-quality DNA [14] is used (not always available), and the possible contamination of tumor tissue with non-neoplastic cells. Furthermore, due to the heterogeneity of methylation patterns between CpG islands, distributions can overlap, making it difficult to define an appropriate cut-off, resulting in a “gray zone”. The latter represents weakly or partially methylated tumors that cannot be assigned to the methylated or unmethylated category [11]. Pyrosequencing is a DNA sequencing technique performed by detecting the nucleotide incorporated by a DNA polymerase through light detection based on a chain reaction due to the release of the pyrophosphate. Using three enzymes—DNA polymerase, ATP sulfurylase, and luciferase—together with the nucleotide (dNTP) in a mixture that is added to the single-stranded DNA, it is possible to evaluate the incorporation of the single nucleotides by measuring the light emitted. This procedure is repeated for each of the four nucleotides until the analysis of the whole single strand is complete.

With PSQ, it is possible to obtain information on the extent of methylation of each CpG site. Various studies have shown that PSQ is a highly reliable method with better performance compared with other techniques [12,15,16]. The reproducibility of the evaluation of the MGMT gene promoter methylation status by PSQ was also evaluated in an inter and intra-laboratory ring trial, demonstrating a high analytical performance of this method [17,18]. Despite the quantitative information on the extent of MGMT promoter methylation status, the reliability and reproducibility of the method, several key questions remain to be answered; in particular, the most important question remains the identification of the best cut-off of MGMT promoter methylation value to differentiate methylated from unmethylated patients. Moreover, another question involves the CpG sites to be analyzed. Although the PSQ has been investigated for CpG sites from 72 to 95 [19], several studies evaluated the CpG sites from 74 to 78, which would seem to be the most informative in prognostic terms [13,20]. CpG 84–87 and 89 seem to have greatest impact on MGMT protein expression; instead, the analysis of CpG 73–81 correlates best with MGMT protein expression [21]. However, the average methylation of the CpGs 74–89 region has shown significant correlation with median OS of glioblastoma patients in three studies [15,22,23].

We performed a large, multicenter, retrospective study to assess the prognostic role of MGMT promoter methylation values by PSQ in glioblastoma.

## 2. Materials and Methods

We retrospectively evaluated all consecutive IDH wild-type glioblastoma patients with an evaluation of MGMT promoter methylation status by PSQ at nine Italian centers from November 2005 to June 2018. The PSQ approach evaluated CpG sites from 75 to 84 of the MGMT gene promoter. Other inclusion criteria were a histologically confirmed diagnosis of glioblastoma (according to WHO 2007 and 2016), Eastern Cooperative Oncology Group (ECOG) PS ≤ 2 before starting oncological treatment, and radio-chemotherapy with temozolomide as a first-line therapy. Patients with carmustine wafer (Gliadel) implantation were excluded, and patients with IDH 1–2 mutated tumors were excluded to avoid interference in the assessment of outcome in this particular patient population.

For each tissue sample, 10-micron sections of paraffin-embedded tumor tissue were provided and processed to isolate the DNA. To avoid any necrotic tissue contamination, through microdissection, only tissue samples of at least 60% of tumor cell were chosen. The DNA was extracted with the QIAamp DNA Mini Kit (Qiagen, Hilden, Germany) following the manufacturer’s instructions. The DNA quality control and yield were assessed by spectrophotometry using Nanodrop machine (Thermo Scientific, Rockford, IL, USA). Two hundred and fifty nanograms of genomic DNA (Positive and Negative Control DNA Set Diatech, Iesi, Italy) were subjected to the MGMT Plus bisulfite conversion kit (Diatech, Iesi, Italy) according to the manufacturer’s instructions. DNA treated with bisulfite was analyzed to determine the methylation status of 10 CpG sites (CpGs 75–84 [24]). The templates for PSQ were amplified through a Rotorgene 6000 with primers that had been biotinylated for template strand (MGMT Plus Kit, Diatech, Iesi, Italy). Twenty microliters of the biotinylated polymerase chain reaction (PCR) products were then immobilized on streptavidincoatedsepharose beads (GE Healthcare, Uppsala, Sweden), and the single-strand DNA templates were analyzed using a PyroMark Q96 system (Diatech, Iesi, Italy). The methylation density for the 10 CpGs analyzed was quantified using the PyroMark Q96 software, and by calculating the average of all 10 methylation sites, the percentage of methylation was obtained for each sample. The samples were run in duplicate. In case of missing data, the analysis is considered inadequate and should be processed on another sample of the lesion.

### Statistical Methods

Continuous data were summarized as medians and interquartile ranges (IQRs), while categorical data were summarized as numbers and percentages. Comparisons between the two groups were performed using the Mann–Whitney U test (continuous data) and Chi-square test (categorical data). The overall survival (OS) was calculated from the date of histological diagnosis to the date of death or last follow-up visit. A time-dependent ROC curve analysis was used to identify a threshold of MGMT promoter methylation for OS. Survival curves were estimated using the Kaplan-Meier method and compared between the groups using the log-rank test. Cox regression models were estimated to assess the effect of MGMT promoter methylation on OS, adjusting for major clinical confounding factors (age, ECOG PS, type of surgery, second surgery, and participating center). The relationship between OS and MGMT promoter methylation was also investigated with Cox regression models, where MGMT promoter methylation was included as a continuous variable, to overcome the limitations due to the dichotomization of a continuous variable. Both a linear relationship (first order polynomial) and a non-linear relationship (restricted cubic splines) were investigated. All tests were two-sided, and a *p*-value of less than 0.05 was considered statistically significant. Statistical analysis was performed using R 4.1 (R Foundation for Statistical Computing, Vienna, Austria) [25].

## 3. Results

### 3.1. Patients

During the study period, 883 patients were treated for histologically confirmed glioblastoma at the nine participating centers. Patient selection is shown in Figure 1. Finally, 591 patients (389 males and 202 females, median age 60 years, IQR 52–67) satisfied the inclusion criteria and were included in the analysis. The patient characteristics are summarized in Table 1. At baseline, 521 patients (88.2%) had an ECOG PS 0–1. All patients underwent surgery; the extent of resection was radical in 343 patients (58.2%) and partial in 246 patients (41.8%) (the information was not available for two patients). All patients received post-surgical treatment with concomitant chemotherapy and subsequent temozolomide according to the Stupp Protocol. Some 98 patients (17.2%) underwent second surgery upon relapse.

### 3.2. Overall Survival

The median follow-up was 15.3 months (IQR10.3–22.9) for 581 patients (follow-up data were not available for 10 patients). At the time of analysis, 460 deaths (77.8%) were recorded. The median OS was 16.2 months (95% CI 15.4 to 18.2), and the OS was 68.9% at 1 year and 29.3% at 2 years (Figure 2).

### 3.3. Prognostic Role of MGMT Promoter Methylation

Overall, the median MGMT promoter methylation value was 4% (IQR 1–21%) (Appendix A). The MGMT promoter methylation showed limited discriminative performance regarding 2-year OS (AUC 0.67; Figure 3A) and the ROC curve suggested a cut-off of 15% for MGMT promoter methylation (sensibility 0.77 and specificity 0.59). The median OS was 25.2 months (95% CI 22.9 to 30.0) in patients with MGMT promoter methylation ≥15% and 14.9 months (95% CI 14.1 to 15.8) in those with MGMT promoter methylation <15%, with a 2-year OS of 51.8% and 18.3%, respectively (*p* < 0.0001) (Figure 3B). When adjusting for a set of clinically relevant confounders (age, ECOG PS, type of surgery, second surgery, and center), MGMT promoter methylation <15% was associated with a greater risk of mortality (HR 2.45, 95% CI 1.98 to 3.05; *p* < 0.0001). The subgroup of patients with MGMT promoter methylation ≥15% included less males (58.9% vs. 69.1%, *p* = 0.02) and had lower ECOG PS (6.3% vs. 14.5%, *p* = 0.02) compared with those with MGMT promoter methylation <15% (Appendix A).

### 3.4. Non-Linear Association between Overall Survival and MGMT Promoter Methylation

To overcome the limitations due to the dichotomization of a continuous variable, the relationship between OS and MGMT promoter methylation was also evaluated with Cox regression models, where MGMT was included as a continuous variable. We investigated both the linear and non-linear relationships and found a non-linear association between OS and methylation of the MGMT promoter (non-linear term: *p* = 0.002) (Figure 4). This finding was confirmed (non-linear term: *p* < 0.0001) when adjusting for major clinical confounding factors (age, ECOG PS, type of surgery, second surgery, and center). The estimated hazard rate was highest around MGMT of 4%, then decreased until MGMT of 40%, and slightly increased for MGMT over 40% (Figure 4B). According to the points describing the shape of the curve (i.e., the points where the curve changed the slope, Figure 4D), the estimated median survival was lowest (14 months), around MGMT of 4%; then increased to 26 months for MGMT of 40%; and slightly decreased around 23 months for MGMT over 40%. When considering MGMT values between the points describing the shape of the curve, patients with MGMT within 0–4% had a median survival of 14.8 months (95% CI 13.8 to 15.8) and a 2-year OS of 18.5%; patients with MGMT within 4–40% had a median survival of 18.9 months (95% CI 16.4 to 21.7) and a 2-year OS of 35.1%; and patients with MGMT within 40–100% had a median survival of 29.9 months (95% CI 23.2 to 43.9) and a 2-year OS of 56.9%.

## 4. Discussion

In this paper, we report the data of a large study regarding the analysis of the MGMT promoter methylation status determined by PSQ. We suggest that a threshold of 15% for methylation of MGMT promoter could have a prognostic role in newly diagnosed IDH wild-type glioblastoma patients, especially on overall survival. MGMT promoter methylation status is an important marker for therapeutic response to temozolomide in glioblastoma patients and has an important predictive role in elderly glioblastoma patients. Two randomized clinical trials supported the importance of cancer therapies stratification based on MGMT gene promoter methylation status in elderly glioblastoma patients. In fact, it has been shown that those with MGMT gene promoter methylation appear to benefit most from temozolomide chemotherapy alone, while those without methylation (unmethylated) have greater benefit from radiotherapy alone [26,27,28]. In the long-term analyses of the randomized phase III trial “NOA-08” [28], the authors concluded that, “to improve OS and event-free survival (EFS), MGMT promoter methylation is a strong predictive biomarker for choice between RT and TMZ and offers unexpectedly favorable long-term outcome with initial TMZ monotherapy”. Moreover, as hypothesized for other biomarkers [29], the methylation status of the MGMT promoter as a possible prognostic factor has also been evaluated in patients with recurrent glioblastoma [30,31]. Overall, a number of open questions remain, mainly concerning the best approach for the assessment of MGMT promoter methylation and the optimal cut-off that significantly correlates with survival [32,33]. Several studies have reported that the best predictive value was obtained by PSQ compared with other techniques such a methylation-specific PCR (MSP) [16,19,34]. With the limitation of the absence of a comparison between MSP and PSQ in the population analyzed in our study, some of the main limitations of MSP are described in the literature, which are partly specific to this technique and partly are common to PSQ: MSP can be performed on samples with DNA extracted from paraffin-embedded [35] and cannot detect heterogeneous patterns of methylation at the primer position, resulting in the possibility of obtaining both false-negative and false-positive results, especially if only low-quality DNA can be extracted [12,13]. The MSP reliability depends often on the availability of reproducible amounts of high-quality DNA, but this is often difficult to achieve in the neurooncological setting [36]. Although severe modifications of this technique have been proposed, there is still no consensus regarding dedicated protocols that allow us to obtain reliable and high-quality results. [14]. Furthermore, the differentiation of methylated or unmethylated tumor status can be difficult if the tumor tissue is “contaminated” with non-neoplastic cells. MSP allows us to identify a cut-off that can however be defined as a technical cut-off: in fact, the value that allows us to differentiate between MGMT methylated and MGMT unmethylated tumor is established as the nadir of quantitative methylation values of the logarithmic curve, deriving from results of many tests. However, these distributions can overlap due to the heterogeneity of methylation across the CpG islands with the consequence that make it difficult to define an appropriate cut-off, resulting in a “gray zone”, and that tumors with partly or weakly methylated cannot be allocated in the cohort of methylated or unmethylated gliomas [11]. In recent years, the use of PSQ has increased compared with quantitative MSP and can be considered the standard diagnostic technique. Noteworthy is the fact that this technique manages to overcome most of the limitations of MSP previously listed. However, a cut-off value that defines the percentage of MGMT promoter methylation that discriminates the methylation status and correlates with survival remains one of the most critical problems. The literature offers differing thresholds resulting from various small studies performed to investigate the prognostic role of MGMT methylation status by PSQ [13]. In our study, we evaluated a large sample of 591 histologically diagnosed glioblastoma patients with homogenous clinical and molecular characteristics (ECOG PS 0–2, IDH1-2 wild-type treated with standard chemoradiotherapy). For all patients, we reported a median OS of 16.2 months. The ROC curve analysis suggested that a threshold of 15% for MGMT promoter methylation may have a prognostic role in such patients, as confirmed after adjusting for relevant clinical confounders.

The MGMT gene promoter comprises 98 CpG sites which are located near the transcription initiation site and is composed of 2 differently methylated regions: CpG 25–50 and 73–90 [24]. In the absence of methylation of these sites, the transcription of the gene takes place regularly, while in the case of methylation of CpG, the transcription of the gene cannot occur with consequential loss of transcription and consequential loss of the enzymatic activity of the protein. The correct selection of the method for analyzing the methylation status of the MGMT gene promoter is of fundamental importance precisely for the methylation heterogeneity of the different CpG sites. While CpG 84–87 and 89 seem to have greatest impacts on MGMT protein expression, instead, the analysis of CpG 73–81 correlates best with MGMT protein expression [21]. However, the average methylation of the CpGs 74–89 region has shown significant correlation with median OS of glioblastoma patients in three studies [15,22,23]. In our study, we analyzed MGMT CpG sites from 75 to 84, and this choice was driven by the best evidence in the literature being reported for the regions 72–83 and 74–89 [13].

The identification of MGMT promoter methylation cut-off that correlated with survival has always been considered one of the most complex and controversial aspects. Several studies have evaluated the methylation status of the MGMT promoter by PSQ and have reported various cut-off values (Table 2 offers a brief summary); most of these studies were retrospective and often analyzed different CpG sites, making it impossible to standardize the various results. Furthermore, although the comparison between our study and these studies cannot be considered accurate for the different CpG sites analyzed, a number of these trial involved small sample sizes and used different types of materials (frozen samples or formalin-fixed paraffin embedded). Our large study suggested that a threshold of 15% for MGMT might have a moderate discriminative performance regarding survival in glioblastoma patients. This threshold lies within the interval of similar cut-offs in the literature, with only a few studies suggesting a cut-off of over 15%; moreover, this type of threshold can be considered reliable only if the same CpGs are analyzed with the PSQ technique. Although identifying a single threshold may facilitate the approach in clinical practice, this appears to be a limitation in terms of clear separation of data into two categories. A recent study with the aim of overcoming this limitation explored a linear association between MGMT promoter methylation and survival [37]. Interestingly, our data suggested that this association may be non-linear, with varying magnitudes of prognostic effect across values of MGMT promoter methylation. Furthermore, the estimated overall survival slightly decreased in the population with MGMT promoter methylation >40% compared with those with 40%. These data could be a starting point for further investigations and studies in this subcategory of patients. This may be useful to personalize risk assessment, but further research is warranted to confirm this finding.

The reader should be aware that the distribution of MGMT values might have influenced the final estimates presented in this study, which therefore requires further investigations to confirm the shape of the non-linear relationship and to achieve more precise estimates. However, we also believe that there is no reason to assume that the relationship between MGMT and OS might be dichotomic or linear, while a non-linear relationship (implying different magnitude of adverse prognostic effect when MGMT varies) seems more reasonable.

The strengths of our study include the large sample size, the inclusion of patients from various Italian neuro-oncology centers (which enhances the generalizability of the findings), and the homogeneity in terms of patient characteristics and analyzed CpG sites. Furthermore, our findings suggest a non-linear relationship between OS and MGMT promoter methylation, which implies a variable magnitude of the effect of decreasing MGMT on prognosis.

However, this study also has a few limitations that should be considered. First, the retrospective design may limit the quality and completeness of the data, although only a few patients were excluded due to missing information on MGMT status. Second, we did not consider the impact of cancer treatments at the time of relapse on OS. Third, to obtain a strong validation of the proposed cut-off for survival, it would have been useful to have an independent validation cohort. Finally, the generalizability of these data should be limited to patients with similar characteristics and with similar CpG sites analyzed.

## 5. Conclusions

This large multicenter study indicated that a threshold of 15% for MGMT might have a moderate discriminative performance regarding survival in IDH wild-type glioblastoma patients treated with radio-chemotherapy with temozolomide as first-line therapy. Moreover, our data suggested a non-linear relationship between OS and MGMT promoter methylation, which implies a varying magnitude of prognostic effect across values of MGMT promoter methylation. Further research is certainly needed in order to confirm these data and to assess the clinical relevance of the methylation status of the individual CpGs.

## Figures and Tables

**Figure 1 cancers-14-02425-f001:**
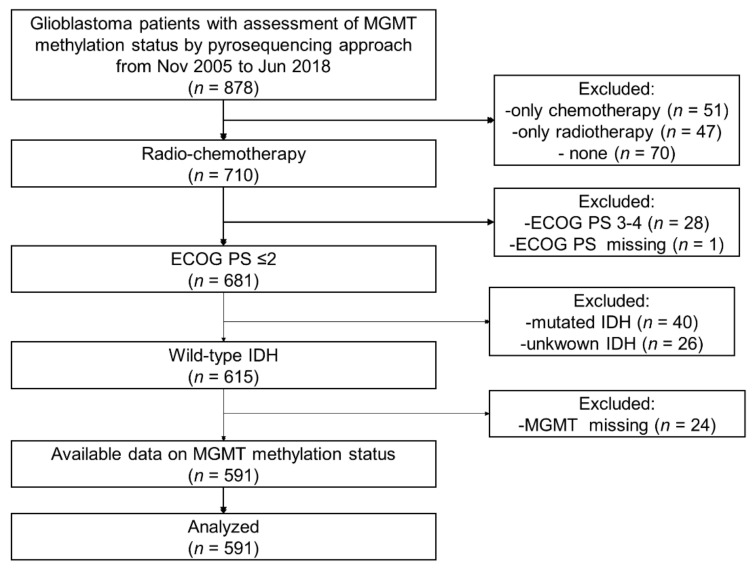
Flow diagram of patient selection.

**Figure 2 cancers-14-02425-f002:**
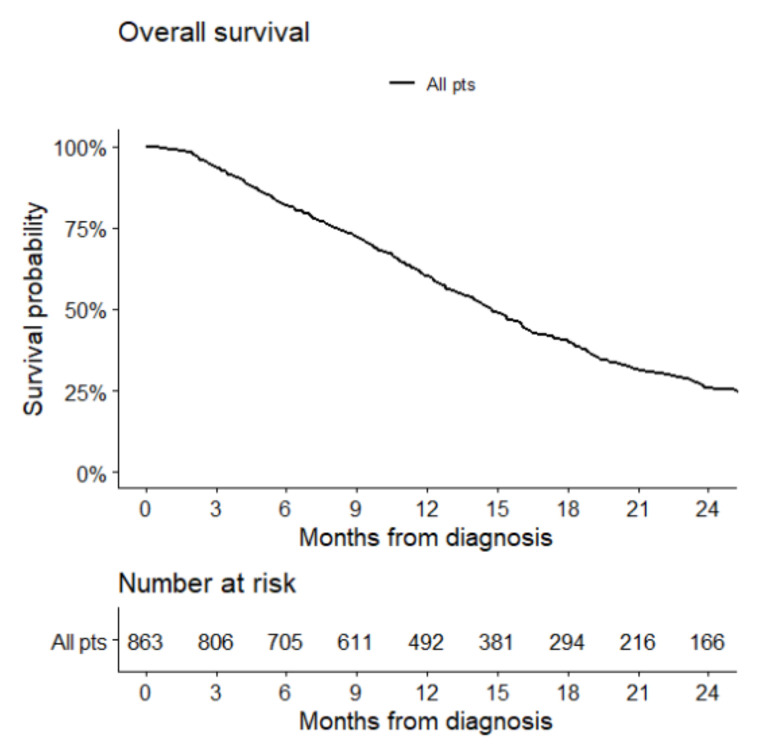
Overall survival of glioblastoma patients with assessment of MGMT promoter methylation status by PSQ approach.

**Figure 3 cancers-14-02425-f003:**
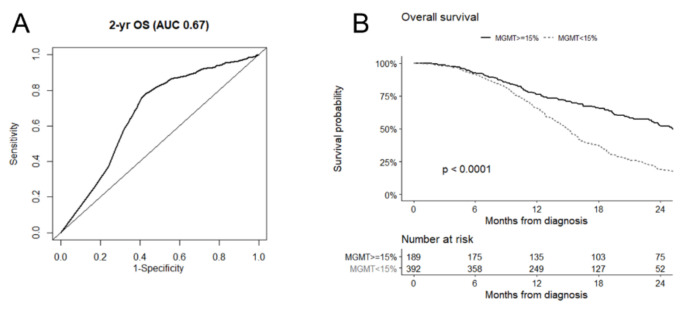
Receiver operating characteristics (ROC) curve of MGMT promoter methylation for 2-year overall survival (OS) in glioblastoma patients (**A**); overall survival for patients with MGMT methylation ≥15% and those with MGMT <15%, according to the cut-off suggested by ROC curve analysis (**B**).

**Figure 4 cancers-14-02425-f004:**
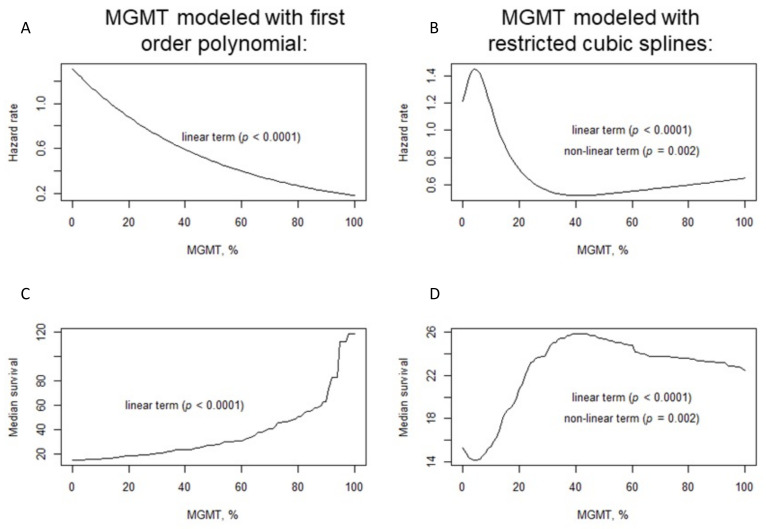
Hazard ratio according to MGMT promoter methylation under the assumption of linear association (**A**) or non-linear association (**B**); median survival according to MGMT promoter methylation under the assumption of linear association (**C**) or non-linear association (**D**).

**Table 1 cancers-14-02425-t001:** Characteristics of glioblastoma patients with assessment of MGMT promoter methylation status by PSQ approach. Data expressed as *n*(%) or ^a^ median (IQR). Data not available in ^b^ 2 and ^c^ 20 patients.

Variable	Summary
*n* patients	591
Age at diagnosis, years ^a^	60 (52–67)
Females Males	202 (34.1) 389 (65.9)
ECOG PS: 0 1 2	255 (43.2) 266 (45.0) 70 (11.8)
Type of surgery: ^b^ Radical Non-radical	343 (58.2) 246 (41.8)
Second surgery	98 (17.2)

Data expressed as *n*(%) or ^a^ median (IQR). Data not available in ^b^ 2 and ^c^ 20 patients.

**Table 2 cancers-14-02425-t002:** Comparison between our study and the largest studies published in the literature regarding methylation status of the MGMT promoter analyzed with PSQ (FFPE: formalin-fixed paraffin embedded).

Study	Samples	Material	Cut-Off (%)	CpGs
Our Study	591	FFPE	15	75–84
Dunn et al. [38]	109	Frozen-FFPE	9, 29	72–83
Rapkins et al. [39]	303	FFPE	9	74–78
Quillien et al. [40]	100	Frozen	8	74–78
Shen et al. [41]	128	Frozen	10	72–83
Mulholland et al. [22]	182	Frozen	10	74–89
Collins et al. [23]	225	FFPE	10	74–89
Reifenberger et al. [42]	166	Frozen	8, 25	74–78
Nguyen et al. [37]	109	FFPE	21	74–78
Li et al. [43]	312	FFPE	5–11	74–81
Hsu et al. [44]	99	FFPE	8	76–79
McDonald et al. [45]	78	FFPE	8	74–78
Xie et al. [46]	43	FFPE	10	74–89

## Data Availability

The data that support the findings of this study are available from the corresponding author upon reasonable request.

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
