# Peer review of "Definition of the Prognostic Role of MGMT Promoter Methylation Value by Pyrosequencing in Newly Diagnosed IDH Wild-Type Glioblastoma Patients Treated with Radiochemotherapy: A Large Multicenter Study"

_cancers, 2022, doi:10.3390/cancers14102425_

Round 1
Reviewer 1 Report
Abstract needs to be more clear and concise with clear abstract conclusion on the O(6)-Methylguanine-DNA-methyltransferase (MGMT) methylation and Radiochemotherapy
Elaborate more on the O(6)-Methylguanine-DNA-methyltransferase (MGMT) activity and repair function of this enzyme
In the Introduction and the Discussion elaborate more on the CpGs and methylation in cancer and elaborate more on the Pyrosequencing
In the Introduction and the Discussion, elaborate more on the Glioblastoma clinics and Radiochemotherapy in details
Fig 3b and Fig 4 Legends need detailed explanation. Explain in details of the firs order polynomial, using mathematical models as well as restricted cubic
Author Response
- Abstract needs to be more clear and concise with clear abstract conclusion on the O(6)-Methylguanine-DNA-methyltransferase (MGMT) methylation and Radiochemotherapy
- We revised the abstract according to the reviewer's recommendations. The revised abstract is shown below:
“Background. MGMT methylation status is a predictive factor for alkylating treatment efficacy for glioblastoma patients and its prognostic role is still unclear. Pyrosequencing is a feasibility method for MGMT methylation testing but its cut-off for survival is unclear. We performed a large, multicenter study to evaluate the association between MGMT methylation value and survival. Methods. From 9 Italian centers, we evaluated glioblastoma patients with assessment of MGMT methylation status by pyrosequencing. Inclusion criteria were:histological diagnosis of IDHwild-type glioblastoma, ECOG-PS≤2, radio-chemotherapy treatment. The relationship between OS and MGMT was investigated with time-dependent ROC curve and Cox regression models with MGMT modeled with first order polynomial or restricted cubic splines. Results. 591 newly diagnosed glioblastoma patients were analyzed. MedianOS was 16.2months. We found a cut-off of 15% for MGMT methylation. The 2year-OS rate was 18.3% and 51.8% for MGMT methylation <and≥15%(p<0.0001). At multivariable analysis, MGMT methylation<15% was associated with impaired survival(HR 2.45;p<0.00001). A non-linear association between MGMT methylation and OS was identified(p=0.002): the medianOS was 14.8months for MGMT in 0-4%, 18.9months for MGMT in 4-40% and 29.9months for MGMT in 40-100%. Conclusions. In our study we identified a threshold of 15% as a strong prognostic value of MGMT promoter methylation by pyrosequencing in newly diagnosed IDHwt glioblastoma patients treated with chemoradiotherapy. Lower values of MGMT promoter methylation were associated with impaired survival and their relationship was non-linear”
- Elaborate more on the O(6)-Methylguanine-DNA-methyltransferase (MGMT) activity and repair function of this enzyme
- We have added in the "Introduction" section a paragraph on the activity of MGMT and repair function of this enzyme. The added paragraph is as follows:
“…to a cysteine residue at its own position 145 (6); it is physiologically involved in the removal of alkyl DNA adducts in the O6 position of guanine, playing a very important role in maintaining the integrity of the genome, also repairing discrepancies and errors during DNA replication and transcription. This repair mechanism prevents cell death, gene mutations, and tumor transformation from occurring due to alkylating agents.”
- In the Introduction and the Discussion elaborate more on the CpGs and methylation in cancer and elaborate more on the Pyrosequencing
- We have added insights into the text about CpGs, methylation in cancer and about pyrosequencing as follows:
“…associated 5’-CpG island of 762 bp including 98 CpG dinucleotides (10), that ecompasses large part of promoter and the first exon. In a situation of normality and in the absence of methylation of the CpG islands, the transcription sites of MGMT together with structures similar to nucleosomes, determine the transcription of the gene regularly. In the case of methylation of the CpGs islands, heterochromatinization and a subsequent rearrangement is determined that does not allow a correct adherence of the nuclesome-like structure, masking the transcription initiation sites, making transcription impossible. Epigenetic alterations such as methylation of CpG islands have always been considered as very early and frequent events in different types of cancer, playing a fundamental role in tumorigenesis and tumor progression. In fact, methylation usually determines the lack of mRNA, with consequent loss of the protein and its enzymatic activity, often favoring tumor development.”
“…that cannot be assigned to the methylated or unmethylated category (11). Pyrosequencing is a DNA sequencing technique performed by detecting the nucleotide incorporated by a DNA polymerase through light detection based on a chain reaction due to the release of the pyrophosphate. Using three enzymes which are DNA polymerase, ATP sulfurylase and luciferase together with the nucleotide (dNTP) in a mixture that is added to the single-stranded DNA it is possible to evaluate by measuring the light emitted, the incorporation of the single nucleotides. This procedure is repeated for each of the 4 nucleotides until the analysis of the whole single strand is complete.”
“…is composed of 2 differently methylated regions: CpG 25-50 and 73-90(24). In the absence of methylation of these sites, the transcription of the gene takes place regularly, while, in the case of methylation of CpG, the transcription of the gene cannot occur with consequent loss of transcription and consequent loss of the enzymatic activity of the protein.”
- In the Introduction and the Discussion, elaborate more on the Glioblastoma clinics and Radiochemotherapy in details
- We have added insights into the text about Glioblastoma clinics and Radiochemotherapy, as follow:
“Glioblastoma (GBM) represent the most common type of primary malignant brain tumor (1). The clinical presentation and symptoms associated with this cancer can be as varied as headache, seizures, and motor and / or sensory neurological deficits.
“chemoradiotherapy treatment and subsequent chemotherapy with maintenance temozolomide (Stupp Protocol); this treatment involves the delivery of radiotherapy (60gy or 40 Gy in 30 or 15 fractions respectively) associated with the chemotherapy treatment with temozolomide at a dosage of 75mg / m2 for the entire duration of the radiation treatment followed by chemotherapy alone with temozolomide at a dosage of 150- 200 mg / m2 for 5 days every 28”
- Fig 3b and Fig 4 Legends need detailed explanation. Explain in details of the firs order polynomial, using mathematical models as well as restricted cubic
Figure 3b displays the survival curves for patients with MGMT methylation ≥15% and those with MGMT <15%, as patients were divided according to the cutoff suggested by ROC curve analysis. We specified this choice in the Results section: “The MGMT promoter methylation showed limited discriminative performance regarding 2-year OS (AUC 0.67; Figure 3A) and the ROC curve suggested a cutoff of 15% for MGMT promoter methylation (sensibility 0.77 and specificity 0.59).”. In the revised manuscript, we improved the legend for figure 3b as: “overall survival for patients with MGMT methylation ≥15% and those with MGMT <15%, according to the cutoff suggested by ROC curve analysis.”.
We acknowledge that the legend of Figure 4 did not convey a clear message to the reader, hence we introduced some changes in the text and in the legend. Figure 4 displays the main estimates from survival analyses (hazard rate and median survival) by MGMT methylation as continuous variable, rather than binary variable. In the revised manuscript, we clarified this aspect: “The relationship between OS and MGMT promoter methylation was also investigated with Cox regression models where MGMT promoter methylation was included as continuous variable, to overcome the limitations due to the dichotomization of a continuous variable.” (Methods). We also specified “Both a linear relationship (first order polynomial) and a non-linear relationship (restricted cubic splines) were investigated.” (Methods), to improve the communication to the reader, as we aimed to underline that the relationship between OS and MGMT might be non-linear and more complex than the usual linear association (i.e. first order polynomial). Following this approach, we left the specification about the type or non-linear association (restricted cubic splines) in the Methods section, while we focused on the core message (non-linearity of the association) in Results section, Figure 4 and Figure 4 legend.
Reviewer 2 Report
This paper investigated a large cohort of glioblastoma patients regarding MGMT promoter methylation status by pyrosequencing, this being an impressive effort. There are though a number of problems that need to be addressed.
Please see comments in your manuscript. Most importantly, MGMT is mainly a predictive factor for alkylating agent treatment and it is debated if it might have a prognostic impact.
When citing another paper please use " ".
That the effect of MGMT would diminish at higher levels (over 40%) would need to be addressed in the discussion.
Please check that it says MGMT promoter methylation everywhere, not just MGMTmethylation.
In the text to figures and tables it does not need to be repeated several times what the time for inclusion was and that there were 9 centers.
In the supplemantary table please calculate MGMT status for men (mMGMT vs uMGMT) and women seperately, so that the proportion within each sex can be determined.
Also some abbriviations and fotnots lack explanation.

Author Response
- Please see comments in your manuscript. Most importantly, MGMT is mainly a predictive factor for alkylating agent treatment, and it is debated if it might have a prognostic impact.
- We have modified the Simple summary and the Abstract, adding the specification requested by the reviewer, as follows:
“MGMT methylation status is a predictive factor for alkylating treatment efficacy for glioblastoma patients and its prognostic role is still unclear”
- When citing another paper please use " "
- We have added " " in the citations of other papers in the text
- That the effect of MGMT would diminish at higher levels (over 40%) would need to be addressed in the discussion
- Even if we do not know at the moment the cause of the reduction of the estimated median survival for values> 40% of the methylation of the MGMT promoter (as reported in figure 4D), we have reported this data in the "Discussion" section, evaluating the possibility that it could act as a starting point for further studies in this subcategory of patients.
“Furthermore, the estimated overall survival slightly decreases in the population with MGMT promoter methylation> 40%, compared to those with 40%. This data could be a starting point for further investigations and studies, in this subcategory of patients.”
- Please check that it says MGMT promoter methylation everywhere, not just MGMTmethylation.
- We have corrected all "MGMT methylation" in "MGMT promoter methylation" in text
- In the text to figures and tables it does not need to be repeated several times what the time for inclusion was and that there were 9 centers.
- We removed from the text of the figures and tables the sentence "at nine Italian center from November 2008 to June 2018".
- In the supplemantary table please calculate MGMT status for men (mMGMT vs uMGMT) and women seperately, so that the proportion within each sex can be determined.
- The supplementary table reports the number of mMGMT males (112), uMGMT males (277), mMGMT females (78) and uMGMT females (124) according to the 15% cutoff of the MGMT status. In addition, we added the number of males and females with MTMT=0% (94 males and 51 females) in the footnote of the supplementary table to provide further information to the reader.
- Also, some abbreviations and fotnots lack explanation.
- We have added the missing explanations of the reported abbreviations and footnotes
Reviewer 3 Report
Methylated patients with methylation less than 15% vs greater than 15% are not well balanced in terms of overall status (PS) this may be a bias when interpreting this cut-off point as valid in its clinical application.
Results could be reported too with the cut-off point at 10% as in most publications to discus with more data because the cut to 15% is advised.
Author Response
- Methylated patients with methylation less than 15% vs greater than 15% are not well balanced in terms of overall status (PS) this may be a bias when interpreting this cut-off point as valid in its clinical application.
- We investigated the relationship between survival and MGMT cutoff adjusting for a set of clinically relevant confounders including PS, and the analysis confirmed that MGMT promoter methylation <15% was associated with a greater risk of mortality. In the text, we reported “When adjusting for a set of clinically relevant confounders (age, ECOG PS, type of surgery, second surgery and center), MGMT promoter methylation <15% was associated with a greater risk of mortality (HR 2.45, 95% CI 1.98 to 3.05; p <0.0001).” (Results).
- Results could be reported too with the cut-off point at 10% as in most publications to discuss with more data because the cut to 15% is advised.
Actually, we showed the results associated with a 15% cutoff for MGMT methylation, but in the end, we advised for using data of MGMT methylation as continuous variable, to avoid the limitations due to the dichotomization of a continuous variable. At first stage, we reported the results of a common approach (dichotomization of a continuous variable and analysis using the identified cutoff), which suggested a cutoff of 15%. At second stage, we investigated the role of MGMT methylation as continuous variable in the analysis, and we found a non-linear association between OS and MGMT, thus implying the suboptimal choice of using a cutoff which dichotomizes a continuous variable
Round 2
Reviewer 2 Report
This paper has been greatly improved after revision, but their is still a need for further revision.
Especially the beginning and discussion of the paper would need minor English language editing.
It is still mainly stated that MGMT methylation status is a prognostic factor, referencing partly to old publications. It is primarily a predictive factor. In the Discussion first paragraph please add the explanations from ref 30-31, that supports MGMT to be prognostic.
In the abstract (and the paper) first a cut-off of 15% is selected (by ROC), then another model for determining the cut-off is presented, that does not clearly compare to the already chosen cut-off of 15%, but divides the cohort into 3 groups. This should also look at the data if the cut-off of 15% would also be included.
Tha manuscript still does not describe how missing data from singel CpG sites for single patients were handled or how many analyses were conducted for each CpG site for each patient. Were the analyses run in dublicat or triplicate?
A table or graph showing the distribution of MGMT values would add valuable information, as with the median stated to be 4%, the numbers of patients above 40% are expected to be very few, which could axplain that the curve is not linear, but sensitive for the lower number together with clinical factors.
Table S1 could be included into the main paper. IQR should be explained
3.4 The sentences starting with ”The estimated…” seem contradictory and need further explanation if correct, as they seem to state different median survival time for the same cohorts, such as those with MGMT 40-100%.
In the Discussion it should be more clearly stated that the cut-off of 15% can only relate to the analyses of exactly the same CpGs by PSQ. As the importance of the different CpGs vary it is expected that also the cut-off for the clinically relevant threshhold will vary, depending on which CpGs are analysed. This is touched on by the authors, but needs further clarification.
Conclusions: It is importatnt to add in the first sentence that the patients received Temozolomide, as the results had been completely different in a cohort that was not treated with an alkylating agent.
Minor revision
Introduction: Sentence ”Playing a very important role”- very should be deleted as also in the sentence ”very early event” and Discussion :”very homogenous…”
2.1 should be ”or last follow up”
Should be Mann-Whitney U test
For figures 2 and 3B, removing the + for each patient on the survival curve would increase readability
Please add p-values to the figures 4A-D
Author Response
Especially the beginning and discussion of the paper would need minor English language editing.
- We have improved English in the beginning and in the discussion
- It is still mainly stated that MGMT methylation status is a prognostic factor, referencing partly to old publications. It is primarily a predictive factor. In the Discussion first paragraph please add the explanations from ref 30-31, that supports MGMT to be prognostic.
- we have corrected and eliminated the term "prognostic" in the “Discussion” section, in order to make the inserted citations appropriate.
- In the abstract (and the paper) first a cut-off of 15% is selected (by ROC), then another model for determining the cut-off is presented, that does not clearly compare to the already chosen cut-off of 15%, but divides the cohort into 3 groups. This should also look at the data if the cut-off of 15% would also be included
- We acknowledge that the clarity of the Abstract could be improved, despite the limitation of 200 words made it difficult to provide full details. The rationale of the flow in the analysis was i) to provide a common analytical approach (identification of a cutoff with ROC technique, which is familiar to most clinicians and medical researchers) at first, then ii) to overcome the limitations of such approach (i.e. the dichotomization of a continuous variable) by investigating the relationship between OS and MGMT as continuous variable, and finally iii) to explore the possibility that the relationship between OS and MGMT might be non-linear. The analysis suggested a non-linear relationship between OS and MGMT, which did not identify any cutoff but described the relationship with a curve (Figures 4B and 4D). As this was difficult to describe with few words in the Abstract (and it is best displayed in Figures 4B and 4D), we reported the median OS for the main sections in which the survival curve could be described (i.e. when a marked change in the slope of the curve could be observed). In the revised manuscript, we rewrote the Abstract to improve clarity for the reader: “ MGMT methylation status is a predictive factor for alkylating treatment efficacy for glioblastoma patients, but its prognostic role is still unclear. We performed a large, multicenter study to evaluate the association between MGMT methylation value and survival. Methods. We evaluated glioblastoma patients with assessment of MGMT methylation status by pyrosequencing from 9 Italian centers. Inclusion criteria were: histological diagnosis of IDHwild-type glio-blastoma, ECOG-PS≤2, radio-chemotherapy treatment. The relationship between OS and MGMT was investigated with time-dependent ROC curve and Cox regression models. Results. 591 newly-diagnosed glioblastoma patients were analyzed. Median OS was 16.2 months. The ROC analysis suggested a cut-off of 15% for MGMT methylation. The 2year-OS was 18.3% and 51.8% for MGMT methylation <and≥15%(p<0.0001). At multivariable analysis, MGMT methylation<15% was associated with impaired survival (p<0.00001). However, we also found a non-linear association between MGMT methylation and OS (p=0.002): median OS was 14.8 months for MGMT in 0-4%, 18.9 months for MGMT in 4-40% and 29.9 months for MGMT in 40-100%. Conclusions. Our findings suggested a non-linear relationship between OS and MGMT promoter methylation, which implies an increasing magnitude of adverse prognostic effect of decreasing MGMT promoter methylation by pyrosequencing in newly diagnosed IDHwild-type glioblastoma patients treated with chemoradiotherapy.”
- Tha manuscript still does not describe how missing data from singel CpG sites for single patients were handled or how many analyses were conducted for each CpG site for each patient. Were the analyses run in dublicat or triplicate?
- Samples were run in duplicate. The PyroMarker CpG software gives a mean methylation value for each 10 CpG site and the total mean of all 10 CpG sites. In case of missing data the analysis is considered inadequate and should be processed on another sample of the lesion (if available)
- A table or graph showing the distribution of MGMT values would add valuable information, as with the median stated to be 4%, the numbers of patients above 40% are expected to be very few, which could axplain that the curve is not linear, but sensitive for the lower number together with clinical factors
- We added the distribution of MGMT values as Supplementary Figure S1, with reference in the revised manuscript (Results and Supplementary Material). We agree that the limited number of patients with high value of MGMT (around 10% of the sample) might have influenced the final estimates from the models, which therefore require further investigations to confirm the shape of the non-linear relationship and achieve more precise estimates. However, we also believe that there is no reason to assume that the relationship between OS and MGMT might be dichotomic or linear, while a non-linear relationship (implying different magnitude of adverse prognostic effect when MGMT varies) seems more reasonable. We added such considerations in the Discussion section: “The reader should be aware that distribution of MGMT values might have influenced the final estimates presented in this study, which therefore require further investigations to confirm the shape of the non-linear relationship and achieve more precise estimates. However, we also believe that there is no reason to assume that the relationship between MGMT and OS might be dichotomic or linear, while a non-linear relationship (implying different magnitude of adverse prognostic effect when MGMT varies) seems more reasonable.”
- Table S1 could be included into the main paper. IQR should be explained
- We changed “IQR” with “interquartile range” in the footnote of Table S1. We would like, if possible, to keep Table S1 in supplementary material for the clarity of the manuscript
- 3.4 The sentences starting with ”The estimated…” seem contradictory and need further explanation if correct, as they seem to state different median survival time for the same cohorts, such as those with MGMT 40-100%
- The first sentence (“The estimated median survival was lowest (14 months) around MGMT of 4%, then increased to 26 months for MGMT of 40% and slightly decreased around 23 months for MGMT over 40%.”) reported the point estimates of survival corresponding to MGMT=4% and MGMT=40% (the points where the curve changed the slope) and the final estimate for MGMT=100%. The second sentence (“Patients with MGMT within 0-4% had a median survival of 14.8 months (95% CI 13.8 to 15.8) and a 2-year OS of 18.5%; patients with MGMT within 4-40% had a median survival of 18.9 months (95% CI 16.4 to 21.7) and a 2-year OS of 35.1%; patients with MGMT within 40-100% had a median survival of 29.9 months (95% CI 23.2 to 43.9) and a 2-year OS of 56.9%.”) summarized the estimated survival for the subgroup of patients within the intervals (i.e. patients with MGMT within 0-4%, those with MGMT within 4-40%, and those with MGMT>40%). We rewrote those sentences to improve clarity for the reader: “According to the points describing the shape of the curve (Figure 4D), the estimated median survival was lowest (14 months) around MGMT of 4%, then increased to 26 months for MGMT of 40% and slightly decreased around 23 months for MGMT over 40%. When considering the subgroups of patients included in the intervals between such points, patients with MGMT within 0-4% had a median survival of 14.8 months (95% CI 13.8 to 15.8) and a 2-year OS of 18.5%; patients with MGMT within 4-40% had a median survival of 18.9 months (95% CI 16.4 to 21.7) and a 2-year OS of 35.1%; patients with MGMT within 40-100% had a median survival of 29.9 months (95% CI 23.2 to 43.9) and a 2-year OS of 56.9%.”
- In the Discussion it should be more clearly stated that the cut-off of 15% can only relate to the analyses of exactly the same CpGs by PSQ. As the importance of the different CpGs vary it is expected that also the cut-off for the clinically relevant threshhold will vary, depending on which CpGs are analysed. This is touched on by the authors, but needs further clarification.
- We specified what the reviewer requested as follows:
"This threshold lies within the interval of similar cut-offs in the literature, with only a few studies suggesting a cut-off of over 15% and moreover, this type of threshold can be considered reliable only if the same CpGs are analyzed with the PSQ technique. "
- Conclusions: It is importatnt to add in the first sentence that the patients received Temozolomide, as the results had been completely different in a cohort that was not treated with an alkylating agent.
- In the "Conclusions" section we specified what the reviewer requested, as follows:
"This large multicenter study indicated that a threshold of 15% for MGMT might have a moderate discriminative performance regarding survival in IDH wild-type glioblastoma patients treated with radio-chemotherapy with temozolomide as first-line therapy. "
MINOR REVISIONS
- Introduction: Sentence ”Playing a very important role”- very should be deleted as also in the sentence ”very early event” and Discussion :”very homogenous…”
- We have removed the word "very" in the "Introduction" and "Discussion" sections as suggested by the reviewer
- 2.1 should be ”or last follow up”
- We corrected with “or last follow up visit”
- Should be Mann-Whitney U test
- We corrected with “Mann-Whitney U test”
- For figures 2 and 3B, removing the + for each patient on the survival curve would increase readability
- We amended Figure 2 and 3b as suggested
- Please add p-values to the figures 4A-D
- We added p-value for linear and non-linear terms in Figure 4A-D as suggested.
Round 3
Reviewer 2 Report
Thank you for the revised manuscript.
I find some suggested revisions only partly implemented, mainly the statement that MGMT methylation status would be mainly prognostic, which apart from in the title is still found throughout the manuscript. Despite its main role as being predicitve is discussed under Discussion.
The addition fo the descriptions af how MGMT works in the introduction is partly overlapping, when describing that it removes methyl groups. It is also in parts difficult to follow.
It is also mentioned that the "Stupp protocol" would include both patients treated to 60Gy and 40Gy, which cannot be considered correct. The 40Gy concomitant radiochemotherapy regim was described by Perry er al. Does this trial include patients treated according to both types of protocol or only 60Gy? If both were used, then this should be clarified in the table of patient characteristics and included in the multivariate analyses.
The description of the non-linear relationship of MGMT methylation status does not seem to be correct, with the statement: increasing magnitude of adverse prognostic effect with decreasing MGMT value. It is, as mentioned mainly not prognostic, but the curve suggest that survival is better below 4% methylation as HR was worst for 4%. The description "when considering subgroups" needs to be further clarified. Do you refer to a multivariate analysis? There the results look linear as survival is best for those with MGMT between 40-100%. The "points" that you explain in your answer to the reviewer need also to be explained for the reader.
Minor:
Summary: Delete "very"
Abstract: Radiochemotherapy "with temozolomide" should be added.
Please add under Methods that Duplicats were run and how missing values were handled
Some minor English editing and careful reading would improve the language further
Under Discussion it looks like you are citing from other publications. It this is the case, please marks these with " ".
Author Response
- I find some suggested revisions only partly implemented, mainly the statement that MGMT methylation status would be mainly prognostic, which apart from in the title is still found throughout the manuscript. Despite its main role as being predicitve is discussed under Discussion.
- We modified the statements of paper referring to the prognostic role of the methylation status of MGMT as follows:
“The “EORTC 26981/22981-NCIC CE.3” study(9) demonstrated that MGMT promoter methylation can be predictor of temozolomide efficacy”
“Morover, as hypothesized for other biomarkers (29), the methylation status of the MGMT promoter as a possible prognostic factor has also been evaluated in patients with recurrent glioblastoma (30,31).”
- The addition fo the descriptions af how MGMT works in the introduction is partly overlapping, when describing that it removes methyl groups. It is also in parts difficult to follow.
- We eliminated the overlapping parts and clarified the transcription mechanism of the MGMT gene as follows:
“O6-methylguanine (O6-MeG)-DNA methyltransferase (MGMT) is a DNA repair protein that catalyzes the transfer of a methyl group from the guanine DNA nucleotide in O6-position to a cysteine residue at its own position 145(6) and playing a important role in maintaining the integrity of the genome, also repairing discrepancies and errors during DNA replication and transcription. This repair mechanism prevents cell death, gene mutations, and tumor transformation from occurring due to alkylating agents. The MGMT gene is located on chromosome band 10q26. It is composed of five exons and has an associated 5’-CpG island of 762 bp including 98 CpG dinucleotides (10), that ecompasses large part of promoter and the first exon. In the absence of methylation of the CpG islands, the transcription sites of MGMT together with structures similar to nucleosomes (nucleosome-like), determine the transcription of the gene regularly. In the case of methylation of the CpGs islands, heterochromatinization is determined that does not allow a correct adherence of the nuclesome-like structure, masking the transcription initiation sites, making transcription impossible. Epigenetic alterations, such as methylation of CpG islands, have always been considered as early and frequent events in different types of cancer, playing a fundamental role in tumorigenesis and tumor progression. In fact, methylation usually determines the lack of mRNA, with consequent loss of the protein and its enzymatic activity, often favoring tumor development.”
- It is also mentioned that the "Stupp protocol" would include both patients treated to 60Gy and 40Gy, which cannot be considered correct. The 40Gy concomitant radiochemotherapy regim was described by Perry er al. Does this trial include patients treated according to both types of protocol or only 60Gy? If both were used, then this should be clarified in the table of patient characteristics and included in the multivariate analyses.
- We have erroneously reported the radiotherapy dosage of 40 Gy in 15 fractions but all patients enrolled in this study performed a treatment with 60 gy in 30 fractions, as per the STUPP protocol. We have eliminated the sentence that incorrectly reported 40 Gy in 15 fractions
- The description of the non-linear relationship of MGMT methylation status does not seem to be correct, with the statement: increasing magnitude of adverse prognostic effect with decreasing MGMT value. It is, as mentioned mainly not prognostic, but the curve suggest that survival is better below 4% methylation as HR was worst for 4%. The description "when considering subgroups" needs to be further clarified. Do you refer to a multivariate analysis? There the results look linear as survival is best for those with MGMT between 40-100%. The "points" that you explain in your answer to the reviewer need also to be explained for the reader.
- In the revised manuscript, we changed "increasing magnitude of adverse prognostic effect of decreasing MGMT promoter methylation" into "varying magnitude of prognostic effect across values of MGMT promoter methylation" (Abstract, Discussion, Conclusions).
In the sentences starting with "When considering the subgroups..", we aimed to offer a simplified description of the non-linear relationship depicted by the curve in fig. 4B-4D to the reader. However, such relationship is easy to show with a figure but harder to describe by using words (as the curve implies a continuous point-by-point variation of the survival). Hence, we calculated the median survival and the estimated 2-year survival for patients with MGMT values that were included in the intervals between the points describing the shape of the curve. In the revised manuscript, we specified: "When considering MGMT values between the points describing the shape of the curve, pa-tients with MGMT within 0-4% had a median survival of 14.8 months (95% CI 13.8 to 15.8) and a 2-year OS of 18.5%; patients with MGMT within 4-40% had a median survival of 18.9 months (95% CI 16.4 to 21.7) and a 2-year OS of 35.1%; patients with MGMT within 40-100% had a median survival of 29.9 months (95% CI 23.2 to 43.9) and a 2-yearOS of 56.9%.".
We also added "According to the points describing the shape of the curve (i.e. the points where the curve changed the slope, Figure 4D),.."
- Summary: Delete "very"
- We have eliminated the word "very"
- Abstract: Radiochemotherapy "with temozolomide" should be added.
- We add “with temozolomide” in abstract
- Please add under Methods that Duplicats were run and how missing values were handled
- We add in the “Methods” section the sentence:
“Samples were run in duplicate. In case of missing data the analysis is considered inadeguate and should be processed on another sample of the lesion.”
- Some minor English editing and careful reading would improve the language further
- We reread and edited the English language in the paper.
- Under Discussion it looks like you are citing from other publications. It this is the case, please marks these with " ".
- We have added " " in the parts where we have mentioned the publications included in the references anyway
“ “to improve OS and event-free survival (EFS), MGMT promoter methylation is a strong predictive biomarker for choice between RT and TMZ and offers unexpectedly fqvorable long-term outcome with initial TMZ monotherapy” ”